# A single n-type semiconducting polymer-based photo-electrochemical transistor

Victor Druet[1], David Ohayon [1], Christopher E. Petoukhoff [2], Yizhou Zhong[1], Nisreen Alshehri[2,3], Anil Koklu[1], Prem D. Nayak[1], Luca Salvigni[1], Latifah Almulla [1], Jokubas Surgailis[1], Sophie Griggs[4], Iain McCulloch [2,4], Frédéric Laquai [2] & Sahika Inal [1] ✉

Conjugated polymer films, which can conduct both ionic and electronic charges, are central to building soft electronic sensors and actuators. Despite the possible interplay between light absorption and the mixed conductivity of these materials in aqueous biological media, no single polymer film has been utilized to create a solar-switchable organic bioelectronic circuit that relies on a fully reversible and redox reaction-free potentiometric photodetection and current modulation. Here we demonstrate that the absorption of light by an electron and cation-transporting polymer film reversibly modulates its electrochemical potential and conductivity in an aqueous electrolyte, which is harnessed to design an n-type photo-electrochemical transistor (n-OPECT). By controlling the intensity of light incident on the n-type polymeric gate electrode, we generate transistor output characteristics that mimic the modulation of the polymeric channel current achieved through gate voltage control. The micron-scale n-OPECT exhibits a high signal-to-noise ratio and an excellent sensitivity to low light intensities. We demonstrate three direct applications of the n-OPECT, i.e., a photoplethysmogram recorder, a light-controlled inverter circuit, and a light-gated artificial synapse, underscoring the suitability of this platform for a myriad of biomedical applications that involve light intensity changes.

Organic mixed ionic-electronic charge conductors (OMIECs) are organic (semi)conductors known for changing their electrical conductivity upon interactions with ionic charges[1]. They can reach various conductance states as a response to small magnitudes of voltage stimulus applied through an aqueous medium. Besides the mixed conductivity, OMIECs possess unique features that have made diverse devices operating in biological media possible[2]. For instance, the ease of biofunctionalization (i.e., immobilization of biorecognition units on film surfaces) led to the development of biochemical sensors[3]. The tunable softness and processability into various architectures resulted

in conducting three-dimensional living tissues[4], while voltage-controlled water uptake and release were used to build mechanical actuators[5].

An often-overlooked characteristic of OMIECs is their ability to absorb light, although most OMIECs are amphiphilic conjugated polymers where exciton generation is coupled to charge transport. The photo-activity of conjugated polymers has been put to use in organic optoelectronic devices such as solar cells[6] and photodiodes[7], which conventionally operate in dry and $O_2$-free environments due to the sensitivity of electronic charge transport to water and ambient

[1]King Abdullah University of Science and Technology (KAUST), Biological and Environmental Science and Engineering Division, Organic Bioelectronics Laboratory, Thuwal 23955-6900, Saudi Arabia. [2]KAUST Solar Center, Physical Science and Engineering Division, Materials Science and Engineering Program, KAUST, Thuwal 23955-6900, Saudi Arabia. [3]Physics and Astronomy Department, College of Sciences, King Saud University, Riyadh 12372, Saudi Arabia. [4]Department of Chemistry, Chemistry Research Laboratory, University of Oxford, Oxford OX1 3TA, UK. ✉e-mail: sahika.inal@kaust.edu.sa

conditions. Some conjugated polymers have been used as photo-catalysts, which use solar energy for green chemical transformations or charge storage in aqueous or organic media[8,9]. It is only over the past decade that their ability to transduce photons into electronic signals has been leveraged at biological interfaces as a wireless, light-based therapeutic tool[10]. Polythiophene-based films[11] or nanoparticles[12] interfacing degenerated rat retinas were shown to act as photovoltaic retinal prostheses that trigger neuronal firing and restore vision upon illumination[12]. Microscale photo-capacitors bear-ing electron and hole transporting organic molecules patterned on an ultrathin self-locking cuff were charged by light and activated the rat sciatic nerve in vivo on-demand at each light pulse[13]. The physical process that an organic semiconductor immersed in an aqueous medium undergoes upon irradiation with light is complex as it may involve electrical (faradaic or capacitive)[14], electrochemical or thermal pathways, or a combination of them[15], while all of these generate a means to interact with biological systems and intervene locally with cellular activity[16,17]. What has been underexplored is how mixed con-duction of OMIECs proceeds when the route of "doping" by light is also activated. The ability to reversibly tune the mixed conductivity of OMIEC films in aqueous electrolytes by applying light instead of vol-tage can open the optoelectronic space for electrochemical device applications that are otherwise not within reach.

Photodetectors convert light input into electrical output, with applications in optical communication, health monitoring, biomedical imaging, and artificial vision[18]. Photodetectors made of organic semi-conductors with adjustable bandgaps for tailored light absorption combine mechanical flexibility, lightweight, low manufacturing cost, and chemical tunability[19]. Most of these devices require a combination of a donor and acceptor material, the so-called bulk heterojunction architecture, to broaden the spectral absorption profile and separate light-induced excitons into free-charge carriers owing to a large interface area and an increased built-in field[20]. Photodetection devices can be in the form of photodiodes and photoconductors (two con-tacts), as well as phototransistors (three contacts). Phototransistors provide an intrinsic amplification, enabling light detection at lower intensities with a broader dynamic range compared to the two contact configurations[21]. A transistor technology known for its record-high amplification performance[22], which can be ideal for photodetection in aqueous media, is the organic electrochemical transistor (OECT) bearing an OMIEC in its channel. A few reports have shown the appli-cation of the OECT as a photodetector. However, these studies do not take advantage of the light sensitivity of OMIECs and instead rely on the incorporation of light-sensitive inorganic materials at the gate terminal, such as gold nanoparticles[21] or quantum dots[23,24]. Integrating such photosensitive materials into the device assembly process introduces an additional step that can be challenging to scale up. Moreover, since these materials are predominantly inorganic, it ham-pers the low-temperature, cost-effective, and solution-processable fabrication of a soft organic-based photodetector. In one device design, a photosensitive polythiophene film is coated onto the gate electrode, which undergoes a light-enhanced oxygen reduction reaction (ORR), thereby modulating the conductivity of poly(3,4-ethylene dioxythiophene) polystyrene sulfonate (PEDOT:PSS) channel[25]. The operation of this OECT is inherently dependent on $O_2$ availability, requires a voltage applied at the gate electrode, and involves faradaic reactions that may not be reversible, leading to potential issues with device lifetime and stability. In fact, all other electrochemical photo-coupled devices rely on a Faradaic photo-detection mechanism, where photogenerated electrons extracted at the gate electrode enable modulation of the channel current. These devices often exhibit high leakage currents and require additional electron donors/acceptors to be added to the electrolyte in order to separate the photogenerated charge. These characteristics make them unsuitable for applications at the biological interface. An alternative

design stacked a photosensitive organic semiconductor blend onto a PEDOT:PSS channel gated through an ionic liquid with a p-doped silicon[26]. In this configuration, however, since the photo-active bulk heterojunction is coupled with the channel, the photodetection does not benefit from the OECT transconductance. Furthermore, these devices use depletion mode OECTs with inherently high OFF currents in dark conditions and high-power demand.

Here we report an n-type organic photoelectrochemical transistor (n-OPECT) based on a single OMIEC as the photoactive material and using light as the sole trigger for operation with no unreversible electrochemical reactions involved. Our platform transduces light into current through a potentiometric mechanism, i.e., the electrochemical potential modulation of an OMIEC-based electrode. The OMIEC is an n-type copolymer based on naphthalene-1,4,5,8-tetracarboxylic-dii-mide-thiophene (NDI-T) backbone that strongly absorbs light in the visible range and stabilizes electrons with cations injected from the electrolyte[27]. We investigate this material's light-responsive (mixed) conductivity in dry and hydrated states through a combination of steady-state and time-resolved spectroscopies, atomic force micro-scopy, and (spectro)electrochemistry techniques. We find that the photoexcited states generated upon light illumination change the electrochemical potential of the film in aqueous media, which then finely modulates its output current. Incorporating the thin polymer in the microscale channel and on the gate electrode that is laterally patterned next to the channel, we build the n-OPECT in one step. The conventional voltage-gated output characteristics are emulated simply by changing the intensity of light that we shine on the device. The light-induced increase in device transconductance is up to 40-fold, sur-passing that of any other reported electrochemical transistors. The photogeneration of current is not accompanied by Faradaic currents, nor does it rely on oxygen or require additional species to be added to the measurement solution. The device is compatible with both aqu-eous and solid-state electrolytes. We leverage the light-included voltage-to-current transduction provided by the n-OPECT circuitry in three diverse applications; a skin-interfacing device recording photo-plethysmography (PPG) signals, a light-activated organic logic circuit, and a retina-inspired neuromorphic transistor. This study offers an avenue for OMIECs, i.e., in-liquid light sensing, by combining optoe-lectronics with electrochemical materials.

## Results and discussion

### Photosensitive electrochemical properties of the n-type film

The n-type polymer (named p($C_6$NDI-T) hereafter) is based on an NDI-T backbone bearing tri-ethylene glycol (EG) side chains anchored to the NDI-unit with a 6-carbon spacer (Fig. 1a). We coated a thin film (ca. 80 nm) from the p($C_6$NDI-T) solution in chloroform on an indium tin oxide (ITO) substrate. First, we collected the UV-VIS absorption spectra of this film immersed in an electrolyte (i.e., PBS, ca. 0.137 M) while doping voltages were applied via an Ag/AgCl reference elec-trode. In its fully undoped state (at +0.4 V vs. Ag/AgCl), p($C_6$NDI-T) displays the characteristic absorption features of NDI-based materials[28], i.e., two peaks at around 350 nm and 620 nm, attributed to the π-π* transition and the intramolecular charge transfer (ICT), respectively (Fig. 1b). When the film is polarized with increasingly negative voltages, the absorption intensity of the ICT peak decreases gradually, and its maximum shifts to 670 nm. Concurrently, new fea-tures appear; one is around 770 nm, followed by another peak around 490 nm, attributed to polaronic species stabilized by electrolyte cations[28,29]. Using a three-electrode configuration (Supplementary Fig. 1a), we recorded the cyclic voltammetry (CV) curve of the film in PBS, which shows increasing reduction currents as the film is subject to negative potentials (Fig. 1c). Two distinct reduction peaks appear at −0.45 V and −0.72 V, accompanied by their oxidation counterparts, indicative of a radical anion and di-anion localized on the NDI unit, respectively[30,31]. When the film is biased beyond the first reduction

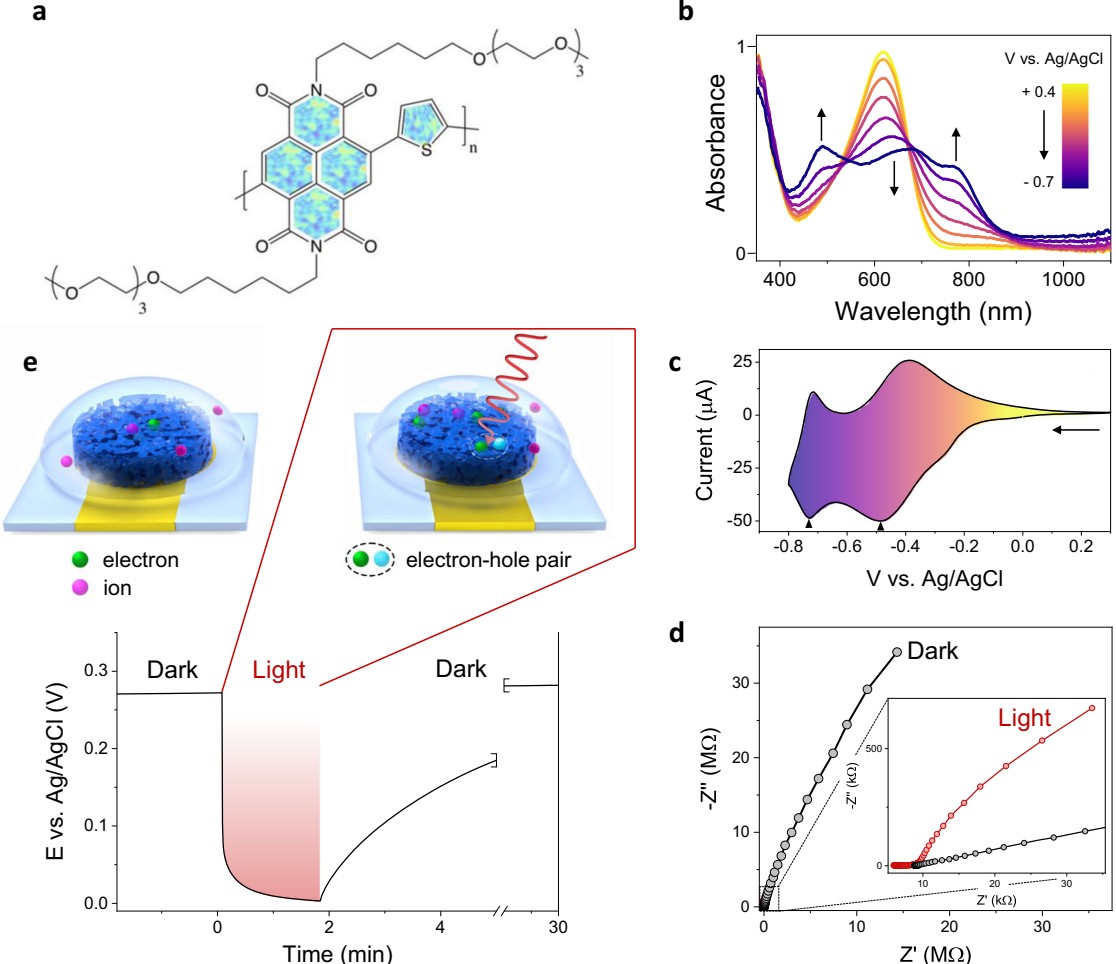

**Fig. 1 | Photoelectrochemical properties of the n-type film probed using a three-electrode setup. a** The chemical structure of p(C₆NDI-T). **b** The evolution of the absorbance spectrum of an ITO-coated p(C₆NDI-T) film during electrochemical doping in PBS. **c** Cyclic voltammetry (CV) curve of a p(C₆NDI-T) film recorded in PBS under dark condition. The film was coated on a circular microelectrode (A = 0.196 mm²). The scan rate was 50 mV/s. The arrow indicates the scan direction, and the reduction peaks are marked. **d** Nyquist plot of the p(C₆NDI-T) electrode in

the dark (black) and when exposed to red light (660 nm, 406 mW/cm²) measured at OCP conditions. The inset highlights the impedance profile when the film is exposed to light. **e** Top: schematic representation of the electrolyte-immersed film interacting with light (red arrow), not to scale. Light forms excitons (mobile electron-hole pairs), and some dissociate into free-charge carriers. Bottom: the change in the OCP of the polymeric electrode upon red light illumination (660 nm, 406 mW/cm²) switched on at t = 0 min for ca. 2 min.

peak potential in PBS (at −0.5 V vs. Ag/AgCl), its electrochemical impedance decreases with the volumetric capacitance reaching ca. 115 F/cm³ (Supplementary Fig. 1b). These results evidence that p(C₆NDI-T) film is a mixed conductor, transporting and coupling electrons and cations, and its conductivity can be reversibly modulated using a broad range of voltages supplied through an aqueous electrolyte by another electrode.

We expect the doping state of all OMIECs to be voltage-dependent; however, for p(C₆NDI-T), we discover that it is also modulated by light. Under red light exposure, the film at open circuit potential (OCP) conditions (i.e., no biasing) shows an almost 500-fold increase in its capacitance and a substantial decrease of the charge transfer resistance (Fig. 1d and Supplementary Fig. 1b, c). The influence of light on the impedance response is much less significant as the film is electrochemically doped (Supplementary Fig. 1b). Since the impedance change with light is maximized when collected at OCP (when the film is not biased), we monitored the OCP of the film with and without light illumination. As the light source was switched on, the OCP, measured at ca. 260 mV vs. Ag/AgCl in the dark, dropped to ca. 3 mV in less than 20 s (Fig. 1e). When we removed the light source, the OCP returned to its original value−although the time it took to generate the light-induced photovoltage drop was much faster than its relaxation to the

dark OCP value, τ extracted from exponential fit ca. 20 s vs. 180 s for the excitation and relaxation, respectively.

## N-type photoresistor

The film's light-responsive electrochemical properties can be generated and probed in other device configurations depending on the output signal targeted. One of these is a resistor configuration. We spin-coated p(C₆NDI-T) on an interdigitated electrode (IDE) array made of gold and measured the current generated by the film when a voltage difference ($V_{IDE}$) was applied between the two electrode terminals (Fig. 2a). In this configuration, the current generated by the film increases when exposed to light (Fig. 2b). The current response is wavelength dependent, as verified by illuminating the film with different LEDs: 455 nm (royal blue), 660 nm (deep red), 730 nm (far red), and 850 nm (infrared), all adjusted to deliver the same power output of 150 mW/cm² (Fig. 2b). The film generates the largest current in response to royal blue (126 times increase in $I_{IDE}$ when going from dark to light, i.e., I_light/I_dark), yet prolonged exposure or stronger light intensity at this wavelength irreversibly degraded the film, which we attribute to the repeated thermal deactivation of excited states (Supplementary Fig. 2a). Irradiation with the deep red LED causes the second largest change in current (I_light/I_dark = 40), higher than the far-

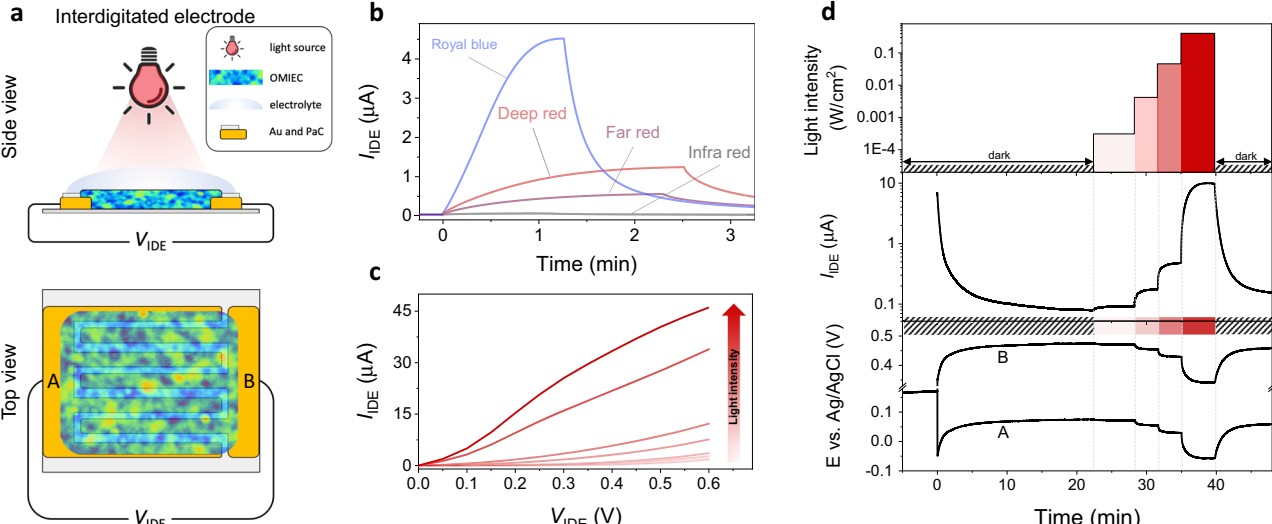

**Fig. 2 | The characteristics of the n-type photoresistor. a** Schematic of the interdigitated electrode (IDE), not to scale. The p(C$_6$NDI-T) film was coated on the IDE (W = 12.8 cm, L = 5 $\mu$m, d = 80 nm) and immersed in PBS. W, L, and d represent width, length, and thickness, respectively. The voltage difference, $V_{IDE}$, was applied between the two contacts (A and B). **b** The response of the IDE current ($I_{IDE}$) to illumination with four independent LEDs with an intensity of 150 mW/cm$^2$ switched on at t = 0 min. $V_{IDE}$ = 0.4 V. LEDs were removed when saturation was reached. **c** *Quasi*-output curve of the IDE where $I_{IDE}$ was measured at various $V_{IDE}$ and during exposure to different light intensities (from 0 to 406 mW/cm$^2$). **d** The real-time changes in $I_{IDE}$ (middle) and the electrochemical potential of the two IDE contacts (bottom) to various light intensities (top). The light intensity application sequence is as follows: dark, 0.3 mW/cm$^2$ (22 min), 4.2 mW/cm$^2$ (28 min), 46 mW/cm$^2$ (31 min), 406 mW/cm$^2$ (35 min), and dark (40 min).

red LED (730 nm, $I_{light}/I_{dark}$ = 20), and infrared light ($I_{light}/I_{dark}$ = 1.5). The current response to the deep red LED is stable and reproducible (Supplementary Fig. 2b). Moreover, the light at this wavelength penetrates through the human skin's epidermis and dermis, suggesting the n-type film's suitability as a skin-interfacing device when employing this light source. Thus, for the rest of the study, we chose the deep red LED (660 nm) as the light stimulus.

Next, we probed whether the film's current is light intensity regulated. We recorded the current output as a function of light intensity (from dark to 406 mW/cm$^2$) while sweeping $V_{IDE}$ from 0 to 0.6 V. Here, the light acts as a gate input to achieve a quasi-output curve (Fig. 2c). The quasi-transfer characteristics recorded at $V_{IDE}$ = 0.4 V show that the IDE current increases sub-linearly with light intensity, with a sensitivity of 173 $\mu$A/(Wcm$^{-2}$) for light intensities lower than 5 mW/cm$^2$ and 50.7 $\mu$A/(Wcm$^{-2}$) for greater intensities (Supplementary Fig. 2c). The device shows an immediate response to light, independent of the intensity of the light (Fig. 2d, middle). The electrochemical potential of device terminals, which we monitored simultaneously, shifted toward more negative potentials as the LED was switched ON, scaled as a function of light intensity (Fig. 2d, bottom). The current profile follows this change in the electrochemical potential of the terminals. Just as the film becomes doped when it is subject to reductive biasing (recall Fig. 1), a brighter illumination pushes the electrochemical potential of the OMIEC film down and brings the film to greater reduction levels, hence doping it further. We do not observe any new electrochemical reactions due to light illumination (no new peaks or loss of features of the CV curve, as shown in Supplementary Fig. 3a), while ORR happens independently of light illumination (Supplementary Fig. 3b). Therefore, the light-controlled electrochemical current and potential changes of the p(C$_6$NDI-T) film do not involve any redox processes as previously identified for other organic semiconductors[14,25]. Moreover, the light-to-current transduction does not require O$_2$ or even the aqueous electrolyte (Supplementary Fig. 3c, d), i.e., the photoresponse is inherent to the p(C$_6$NDI-T) backbone. However, the highest current response to light (i.e., $I_{light}/I_{dark}$) is recorded when the film is swollen with electrolyte (and in the absence of O$_2$ due to large ORR-free

currents extracted), suggesting that the water permeability and ionic conductivity of the p(C$_6$NDI-T) film enhances its photoresponse. Oligoether side chains have been reported to enhance the lifetime of the electron polarons of organic semiconductor films, attributed to an increased local dielectric aiding in charge stabilization[32]. Quartz crystal microbalance with dissipation monitoring (QCM-D) measurements revealed that upon immersion in 0.1 M aqueous electrolyte solution, the p(C$_6$NDI-T) film increased its mass by ca. 31% (Supplementary Fig. 4). We hypothesize that the high dielectric constant of water ($\varepsilon_{water} \sim 80$), compared to the low dielectric constant of organic material ($\varepsilon_{OMIEC} \sim 2-4$), contributes to the amplified photoresponse by reducing the exciton binding energy or retarding the charge recombination mechanisms through the water-swollen film. To investigate this further, we conducted photocurrent experiments with films swollen in solvents with different dielectric constants. We observed drastic differences in the photoresponse, in line with this hypothesis (Supplementary Fig. 5). The water-ingested film provides an environment that better shields coulombic fields between photogenerated charges[32], hence our single donor-acceptor polymer is able to generate drastic voltage changes upon illumination.

## Photophysical properties of the n-type film

Since photocurrent can be generated in dry as well as de-oxygenated environments, we sought to understand the mechanism behind the photocurrent response of the film using well-established steady-state and time-resolved spectroscopy methods. Although we found that the light response is not conditional on the presence of O$_2$ or water, we first verified whether the film is light-sensitive in dry conditions with no bias (no $V_{IDE}$). We probed the film surface morphology and potential profile using amplitude modulation Kelvin probe force microscopy (AM-KPFM). While the surface morphology shows no changes with the light, we recorded a 50 mV drop in the surface potential, recovering each time the light pulse is turned OFF and appearing once it is ON (Fig. 3a). We estimated the work function of the film as 4 and 4.05 eV in the dark and under light, respectively. Having verified the light sensitivity of the intrinsic (unbiased) and dry

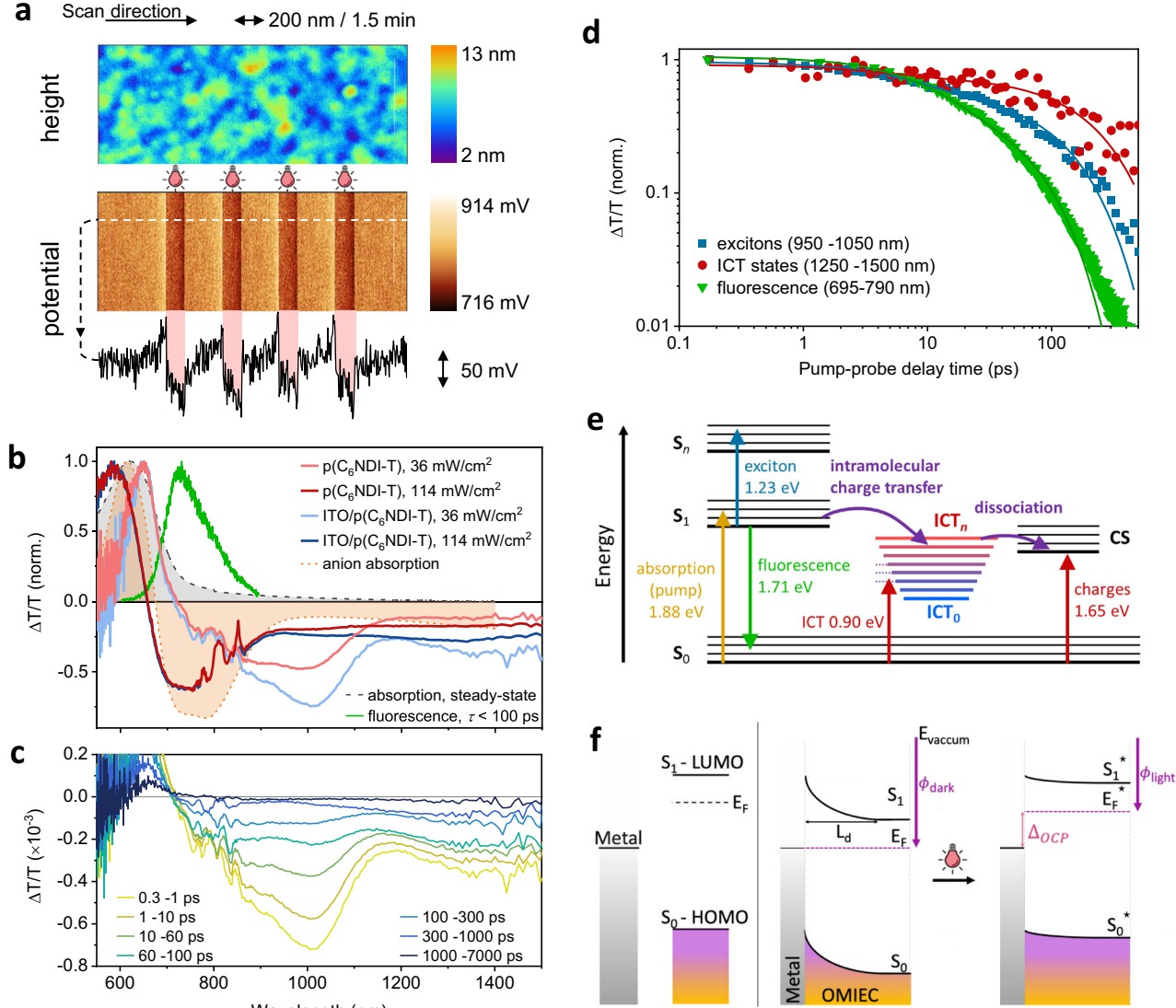

**Fig. 3 | The photophysical characterization of the dry film and the mechanism for the light-induced charge generation. a** In situ AM-KPFM scan of the film surface during exposure to 60 s long light pulses (5 mW/cm$^2$). Top: height data, middle: corresponding surface potential, bottom: potential profile along the dotted line of the middle panel. Light pulse intervals are highlighted in red. The substrate was ITO. **b** TA spectra of intrinsic p(C$_6$NDI-T) film (red) and the ITO/ p(C$_6$NDI-T) bilayer (blue) at different excitation irradiances at a pump-probe time delay of 0.3–1 ps. Overlaid are the DIA anion spectra (orange), steady-state absorption (gray), and time-resolved fluorescence spectra integrated over the first 100 ps (green). Note that the region around 750–850 nm in the TA spectra is impacted by the 800 nm white light seed scattering (Supplementary Fig. 9). **c** TA spectra of the ITO/p(C$_6$NDI-T) bilayer at different pump-probe delay times for irradiance of 36 mW/cm$^2$ (i.e., fluence of 24 μJ/cm$^2$). **d** Picosecond-nanosecond kinetics of the ITO/p(C$_6$NDI-T) interface, compared with the time-resolved fluorescence kinetics. Overlaid are double exponential decay fits (Supplementary Table 1). **e** Energy diagram describing the formation of an ICT complex at ultrafast timescales and the ICT dissociation process into fully charge-separated (CS) states. Vertical upward transitions represent the pump and probe wavelengths of the different photo-excited species. **f** The proposed mechanistic view of the light-induced surface photovoltage. Energy band representation of the metal and the OMIEC before being in contact (left). S$_1$, S$_0$, and E$_F$ stand for the first excited state, the ground state, and the Fermi level, respectively. Fermi level pinning before illumination and the induced band flattening upon illumination (right). Depletion layer and surface photovoltage change are denoted by L$_d$ and $\Delta_{OCP}$, respectively.

p(C$_6$NDI-T) films, we proceeded with our time-resolved spectroscopy measurements.

Transient absorption (TA) spectroscopy is a powerful technique that probes the spectro-dynamics of photoexcited states, allowing for the elucidation of charge generation, transfer, and recombination processes (see "Methods")[33,34]. Using TA spectroscopy, we can differentiate between excitons and charge carriers based on their specific spectral signatures and lifetimes. The TA spectra of the p(C$_6$NDI-T) film showed a positive change in transmission ($\Delta T/T$) signal in the range of 620–670 nm, corresponding to its ground state bleaching (Fig. 3b). At longer wavelengths, the intrinsic p(C$_6$NDI-T) film showed a dominant, broad photoinduced absorption (PIA), or negative $\Delta T/T$ signal, at

900–1100 nm, and a much weaker PIA signal at 1250–1500 nm. We attribute the signal at 900–1100 nm to the PIA of excitons in the neat p(C$_6$NDI-T) film (Supplementary Fig. 6). To check for the presence of free-charge carriers in the dry p(C$_6$NDI-T) films, we chemically doped the film with cobaltocene to determine the anion or doped polymer film spectrum (see "Methods"). The doping-induced absorption (DIA) spectrum (Fig. 3b) showed an increased $\Delta T/T$ at the peak ICT steady-state absorption band (580–650 nm) and a broad negative $\Delta T/T$ signal from 690–850 nm, which correlated well with the electrochemically doped polaron features observed (Fig. 1b). Since the TA signal from the intrinsic p(C$_6$NDI-T) film was weak in the range of 690–850 nm, we postulate that the dry films produce only a small number of free

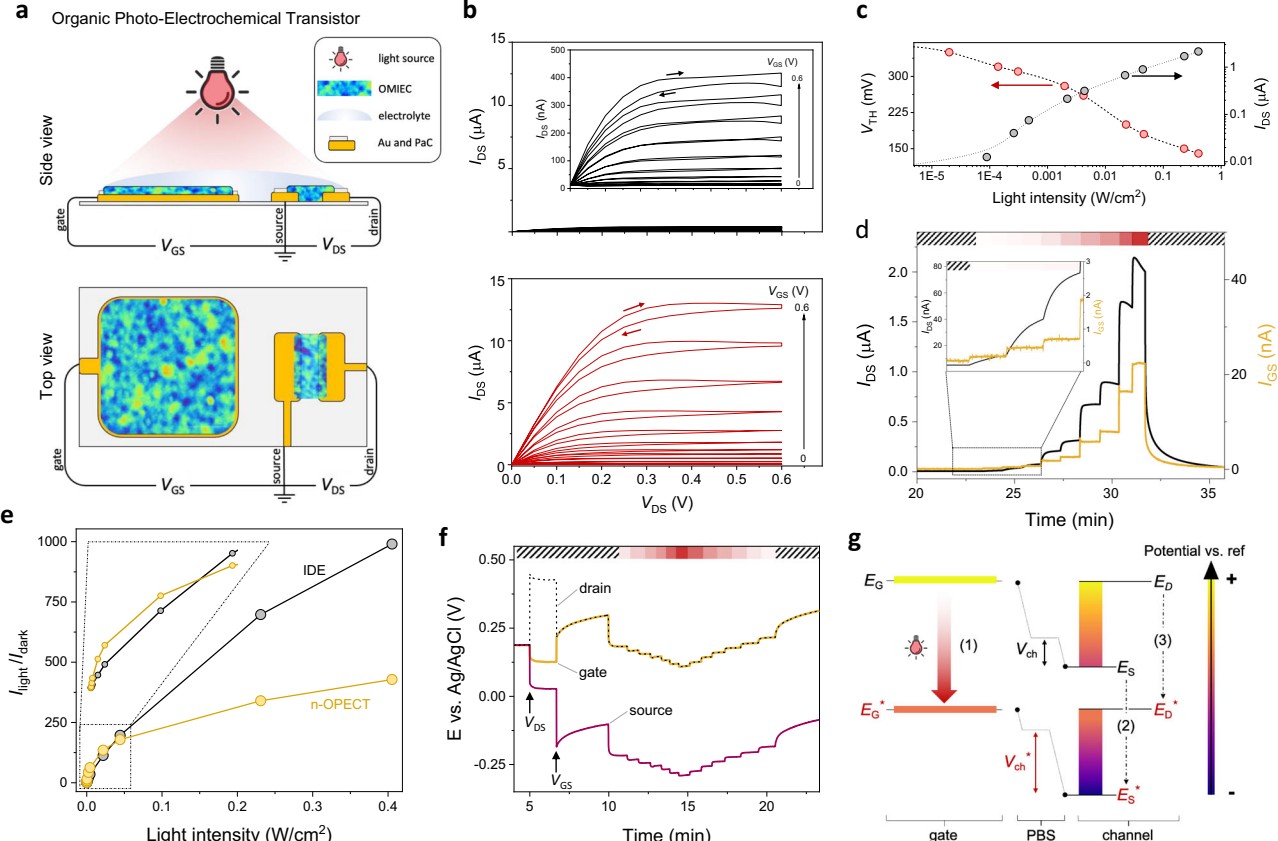

**Fig. 4 | Characteristics of the n-type organic photoelectrochemical transistor.**
**a** Schematic of the n-OPECT, not to scale. The source contact is the electrical ground. The p($C_6$NDI-T) film is coated on the channel (W = 100 μm, L = 10 μm, d = 80 nm) and the gate electrode (500 × 500 μm, d = 80 nm), both immersed in PBS. **b** Output characteristics in the dark (top) and under light (406 mW/cm², bottom). The inset in the top panel shows the at-scale data in the dark. Right-hand side arrows indicate the increase in $V_{GS}$. **c** The threshold voltage ($V_{TH}$, left) and channel current response ($I_{DS}$, right, monitored at $V_{DS} = V_{GS} = 0.4$ V) as a function of light intensity. **d** Current-time profile upon increasing the light intensity from 0 to 406 mW/cm². $V_{DS} = V_{GS} = 0.4$ V. **e** $I_{light}/I_{dark}$ as a function of light intensity for the n-OPECT and the IDE platform. The inset focuses on the response to light intensities below 50 mW/cm². **f** Electrochemical potentials of the n-OPECT terminals during

device operation. $V_{DS} = 0.4$ V is applied at t = 5 min, followed by $V_{GS} = 0.4$ V at t = 7 min. LED is switched on at t = 10 min, with a stepwise increase in its intensity until t = 15 min and a stepwise decrease until t = 21 min, which is when the LED is switched off. **g** The light-induced change in the electrochemical potentials of the device terminals ($E_G$ for gate, $E_D$ for drain, and $E_S$ for source) with respect to the Ag/AgCl reference electrode. The sequence of changes during the light exposure is as follows: (1) $E_G$ shifts downward to $E_G^*$ toward a greater reductive state (more negative vs. Ag/AgCl). (2) $E_S$ shifts to $E_S^*$ to maintain the magnitude of the $V_{GS}$. (3) $E_D$ shifts to $E_D^*$ to maintain $V_{DS}$. The more negative state brings the channel to a more doped state, but the voltage drop at the channel-electrolyte interface ($V_{ch}$) increases due to a more capacitive gate.

charges. At higher excitation irradiance, there was a very pronounced feature in the range of 690–850 nm (Fig. 3b) due to the presence of charges formed from ultrafast photoexcitation to high-energy states, leading to efficient exciton dissociation[35,36] (Supplementary Fig. 7). This ultrafast exciton dissociation process is typically several orders of magnitude lower under continuous wave illumination than femtosecond pulsed excitation[37].

Since our bioelectronics platforms comprised a high-work function metal (gold) as a current collector, we investigated the impact of an ITO (high-work function transparent conducting oxide) interface on the photoexcited species formation within p($C_6$NDI-T). With ITO underneath, the PIA signals of p($C_6$NDI-T) at 950–1050 nm and 1250–1500 nm were enhanced relative to the intrinsic polymer by factors of 1.6 and 3, respectively. Much of the exciton PIA signal around 950–1050 nm decayed in under 300 ps, whereas there was a pronounced PIA signal in the range of 1250–1500 nm up to several nanoseconds (Fig. 3c). The amplitude-averaged lifetime of the 1250–1500 nm state was 186 ps, whereas that of the 950–1050 nm state was 81 ps, suggesting that the two signals arose from different species (Fig. 3d). Compared with the fluorescence lifetime (amplitude-averaged, 37 ps), which gives the lifetime of purely excitonic species, both

states were much longer, demonstrating that neither state was purely excitonic. The low fluorescence lifetime of intrinsic p($C_6$NDI-T) (Supplementary Fig. 6) suggests that the transitions within the film are dominated by non-radiative pathways, such as charge separation or the formation of charge transfer states[38]. We thus assign the 1250–1500 nm species to PIA of ICT states formed at the moment of photoexcitation due to its longer lifetime, more pronounced intensity in the presence of ITO, and its similar spectral characteristics to the high-irradiance TA and DIA spectra in the 1200–1400 nm range. This ICT state is an intermediate between that of a pure Frenkel exciton and fully dissociated charges (Fig. 3e) and possesses both exciton and charge-like characteristics. The species becomes more pronounced upon resonant excitation of the p($C_6$NDI-T) film (Supplementary Fig. S8). There was likely significant mixing between the exciton and ICT states, which gave rise to the short exciton fluorescence lifetime, instantaneous formation of ICT states, and the hybrid exciton-charge-like nature of the ICT state spectro-dynamics[39–41]. The different PIA signals and charge transfer and dissociation pathways are summarized schematically in Fig. 3e.

Although not a significant number of free charges formed intrinsically in the dry film, the presence of the ICT states that formed

instantaneously at the moment of photo-excitation, particularly at the ITO/p(C$_6$NDI-T) interface, could explain the photo-sensitivity of the electrochemical properties of the OMIEC devices. When interfacing the n-type p(C$_6$NDI-T) on high-work function metallic electrodes, such as Au or ITO, a Schottky barrier is formed, giving rise to band bending and a built-in potential at the interface (Fig. 3f)[42]. Under illumination, the electron *quasi*-Fermi-level shifts upward, owing to the presence of a depletion layer (L$_d$) at the metal-OMIEC interface, which rapidly sweeps holes toward the metal and electrons toward the OMIEC[43], thereby increasing the carrier photogeneration rate. This *quasi*-Fermi level splitting gives rise to the photovoltage ($\Delta_{OCP}$) of the junction, in agreement with the 50 meV shift in surface potential that we observed in the KPFM scans (recall Fig. 3a). We propose that the instantaneous formation of ICT states in the OMIEC film facilitates charge dissociation at interfaces, such as the ITO-OMIEC or electrolyte-OMIEC interface. All photo-excited states described above certainly account for the OCP downward shift observed in electrolytes (recall Fig. 1e) and eventually lead to the increased photoconductivity observed in the OMIEC-based devices.

## N-type organic photoelectrochemical transistor (n-OPECT)

The n-type IDE converts light input into a current output; however, an electrode is a passive element (no amplification), its response to light is slow (in the order of 30 s, Fig. 2d and Supplementary Fig. S3c), and decreasing its size (to improve its speed and spatial resolution) would lower its light detection performance. An OECT, on the other hand, is an amplifying transducer that can be an excellent miniaturized photodetector compatible with aqueous media. We fabricated a microscale channel (100 × 10 μm) and an Au gate electrode (0.25 mm$^2$) side-by-side patterned with the n-type film, as illustrated in Fig. 4a. A positive gate voltage ($V_{GS}$) applied through the n-type gate pushes electrolyte cations into the channel that compensate for the electrons injected from the source. With an increase in $V_{GS}$, the channel generates more current ($I_{DS}$) saturating at high source-drain voltages ($V_{DS}$), exhibiting the typical characteristics of an enhancement mode transistor (Fig. 4b-top). When the device is illuminated with the LED at 660 nm (406 mW/cm$^2$), we record a dramatic increase in $I_{DS}$ in the output characteristics (Fig. 4b-bottom). For instance, $I_{DS}$, which is only 400 nA in dark conditions (at $V_{GS} = V_{DS} = 0.6$ V), reaches up to 13 μA under light irradiation. The light also shifts the threshold voltage ($V_{TH}$) from 350 mV to 140 mV at 406 mW/cm$^2$ (Fig. 4c, Supplementary Fig. 10a). The $V_{TH}$ shift can be controlled with light intensity, with a semi-logarithmic dependence in line with the surface photo-voltage model, similar to a Schottky diode[42,43].

Figure 4d depicts the real-time changes in $I_{DS}$ as the light intensity increases. At $V_{DS} = 0.4$ V and $V_{GS} = 0.4$ V, the channel current exhibits a ca. 430-fold increase from the dark state to illumination at 405 mW cm$^{-2}$. Under light exposure, the device transconductance experiences a significant boost ($g_{m, dark} = 20$ mS/cm vs. $g_{m, light} = 775$ mS/cm), suggesting that light can serve as a secondary stimulus, enabling high gain for the detection of very low concentrations of biochemicals and resolution of weak physiological signals. The $I_{DS}$ values plotted as a function of light intensity yield the calibration plot shown in Supplementary Fig. 10b, with a maximum sensitivity of 151 μA/(Wcm$^{-2}$) achieved for intensities below 0.5 mW/cm$^2$. In Fig. 4e, we compare the $I_{light}/I_{dark}$ ratio of the n-OPECT and the IDE as a function of light intensity to highlight the advantages of each system as a photodetector. The n-OPECT exhibits higher sensitivity toward lower light intensities (see inset). The lowest limit of detection for the n-OPECT is 0.7 μW/cm$^2$, whereas it is 30 μW/cm$^2$ for the IDE. As the light intensity increases beyond 4 mW/cm$^2$, the $I_{light}/I_{dark}$ changes less for the OECT. Note, however, that the IDE has an active, light-exposed area 16 times larger than that of the OECT (4 mm$^2$ vs. 0.25 mm$^2$). Moreover, thanks to the miniaturized geometry of the channel, the time required for the photocurrent to stabilize is 10 times faster for the OPECT

($\tau_{OPECT} \sim 3.3$ s vs. $\tau_{IDE} \sim 36$ s) when analyzed for the same intensity and length of the light pulse (Supplementary Fig. 11). We compare the figure of merits of our device with other OPECTs in Supplementary Table 2, demonstrating its superior performance.

In Fig. 4d, we also observe an increase in the gate current ($I_{GS}$) upon illumination in addition to the channel current. The current generated at the lowest intensity illumination (0.09 mW/cm$^2$) is 375-fold and 7.5-fold larger than the noise (dark) current for the $I_{DS}$ and $I_{GS}$, respectively, highlighting the significant amplification and high signal-to-noise light detection granted by the transistor circuitry[44] However, the change in $I_{GS}$ with light is not attributed to photogenerated current. The $I_{GS}$ in the n-OECT arises from ORR, and as ORR current increases with light (recall Supplementary Fig. 3a, b), so does $I_{GS}$. When we monitor the channel current by eliminating ORR, we observe a substantial photocurrent response (similar to the IDE currents shown in Supplementary Fig. 3c), while $I_{GS}$ remains unchanged (Supplementary Fig. 12). $I_{GS}$ not scaling the change in $I_{DS}$ demonstrates the non-Faradaic nature of our photodetection, unlike other reports that rely on a light-responsive OECT current[25,45,46].

To understand the device operating mechanism, we monitored the change in the distribution of the electrochemical potentials at the two electrolyte interfaces under light irradiation and electrochemical doping (Fig. 4f). When a constant $V_{DS}$ is applied, the electrochemical potential of the source terminal decreases with respect to the drain, with the difference approximating the applied $V_{DS}$ (0.4 V). A $V_{GS}$ of 0.4 V drops the source potential down to -0.1 V vs. Ag/AgCl, with the gate terminal stabilizing at 0.3 V vs. Ag/AgCl. A stepwise increase in the light intensity drops the potentials of all terminals, which is completely reversible with a decrease in the light intensity or as it is removed. When going from dark to light, the more negative potential of the gate electrode pulls the source to higher reduction potentials (from -0.1 V in the dark to -0.3 V under light, vs. Ag/AgCl) to maintain the $V_{GS}$ applied by the source-measure unit. As such, the channel becomes more conducting without increasing the $V_{GS}$. Simultaneously, the polymeric gate electrode polarized at lower positive potentials under illumination (from 0.3 V in the dark to 0.1 V under light, vs. Ag/AgCl) attains a higher capacitance, which leads to more effective gating[47]. The light-induced OECT operation is illustrated in Fig. 4g. Overall, light emulates the effect of gate biasing by bringing the channel to a more doped state (steps (1), (2), and (3)) and allowing a larger potential drop at the channel/electrolyte interface ($V^*_{ch} > V_{ch}$).

Since both electrolyte interfaces of the n-OPECT are made of the p(C$_6$NDI-T) film, it is unclear which terminal drives the photoresponse. We, therefore, selectively illuminated either the gate or the channel of the same device and recorded the corresponding photocurrent. Illuminating only the gate electrode increases $I_{DS}$ 25-fold compared to its dark value (Supplementary Fig. 13a, left). However, when only the channel is illuminated, the change in $I_{DS}$ is much lower (a 1.07-fold increase compared to the dark value) (Supplementary Fig. 13a, right). On the other hand, when the n-type channel is gated with an external Ag/AgCl electrode, the current is better modulated, but $I_{light}/I_{dark}$ remains lower than 1.5 regardless of the potentials applied (Supplementary Fig. 13b). OECTs amplify potentiometric changes at the gate electrode, making it advantageous to have the sensing event occur at the gate electrode[48]. In our device, the additional amplification is derived from the fact that the OMIEC at the gate electrode exhibits higher capacitance (or is more electrochemically doped) under light conditions (recall Fig. 1d and Supplementary Fig. 1a), resulting in an additional potential drop in the channel-electrolyte interface. This mode of operation sets the n-OPECT apart from other electrochemical photodetectors and gives it superior performance (as shown in Supplementary Table 2). Taken together, these results demonstrate that the photo-activity of the n-type film at the gate electrode primarily governs the n-OPECT operation, suggesting the potential for enhancing photodetection performance through deliberate material

selection in the channel and optimization of device geometry to maximize the voltage drop at the channel.

## Applications of the n-OPECT

PPG is a non-invasive optical measurement method for monitoring heart rate and blood oxygenation levels. The PPG setup contains a light source and a photodetector which measures the changes in the intensity of light transmitted or reflected from the tissue, directly correlated to the volumetric changes in blood flow from the peripheral circulation. We used the n-OPECT to monitor in real-time the variation in the intensity of light transmitted from the index finger of a volunteer in transmission mode. The human tissue was placed between the light source and the gate electrode (p(C$_6$NDI-T)-coated ITO) (Fig. 5a-i). Instead of using an aqueous electrolyte, we connected the gate to the channel through an agarose-based ion-bearing hydrogel. Using the hydrogel allowed us to place the device upside down so that the finger directly contacted the backside of the ITO substrate that carries the gate, granting stable signal acquisition. The transmitted light (150 mW/cm$^2$ before transmission, 660 nm) is captured by the n-OPECT and converted into a PPG waveform that reflects the pulsatility of the circulatory system (Fig. 5a-iii). Although the amplitude of the waveform acquired from the $I_{DS}$ is only ca. 400 pA, we can identify the PPG characteristics, including the systolic and diastolic peaks (represented as S and D, respectively, i.e., the fundamental and secondary peaks in PPG cycles, shown in the inset of Fig. 5a-ii). This level of signal resolution, granted by the transistor circuitry, enables the determination and extraction of important diagnostic information, such as pulse rate, blood pressure level, and vascular aging degree. Figure 5a-iv shows the PPG amplitude versus frequency spectrum (beats per minute, bpm, as the unit) processed through Fast Fourier Transform (FFT) analysis, indicating an overall pulse rate of 63 bpm with the fundamental peak in the frequency domain. Moreover, the n-OPECT as a PPG sensor has very low power demand. We measured an average power consumption of 0.63 μW. This value is 500 times lower than the power demand of the commercially available reference PPG device we used to validate the accuracy of our device (30 μW).

The n-OPECT is an ideal element for building a light-gated inverter logic circuit. In a traditional inverter circuitry, the output voltage is modulated by the gate voltage input, acting as a voltage-to-voltage converter. We connected the p(C$_6$NDI-T) channel in series with a p-type OMIEC channel, i.e., namely p(g0T2-g6T2)[49], and immersed both films in PBS that share a common p(C$_6$NDI-T)-coated gate electrode (Fig. 5b-i). The dimensions of the two channels were selected such that the two OECTs display matching characteristics (Fig. 5b-ii). The middle point between the two channels is the output voltage ($V_{OUT}$) of the logic circuit, which switches between the drive voltage, $V_{DD}$ (p-channel ON/n-channel OFF), and the ground, GND (p-channel OFF/n-channel ON). Additionally, a gate voltage ($V_{IN}$) is applied on both channels. The logic circuit's input-output and corresponding gain ($V_{OUT}/V_{IN}$) characteristics are displayed in Fig. 5b-iii, top and bottom panels, respectively. In Fig. 5b-ii, we observe a slow transition (low gain) of $V_{OUT}$ from 0.15 V to 0 V when using the gate in dark conditions. Upon illumination, the inverter characteristics match the one when using an Ag/AgCl as the gate electrode, displaying a much sharper ON/OFF transition (higher gain) centered around $V_{IN}$ = 0.12 V with a gain of 2. Having the characteristics of the two conditions (dark and light) in mind, we keep $V_{IN}$ = 0.175 V constant to switch ON and OFF the inverter solely with light. We show in Fig. 5b-iv that upon light exposure (t = 0 s), $V_{OUT}$ switches from ON (ca. 0.15 V) to OFF (ca. 0 V) within a few seconds, while turning the light off (at t = 20 s) makes the logic circuit to return to its ON state within 100 s. This device is a demonstration of an OECT-based inverter circuit controlled by light illumination.

The third application of the n-OPECT is an artificial synapse. Neurobiological processes involve ionic fluxes and biochemicals that are sent from one neuron to another through a synaptic junction (Fig. 5c-i). The connection strength between neurons changes over time. This synaptic plasticity can be short (ms to min) or long (>10$^3$ s) and is at the basis of any synaptic function responsible for, for instance, learning or memory[50]. We developed a synaptic n-OPECT where $V_{GS}$ and the light stimuli act as the presynaptic stimuli (S$_{pre}$), and $I_{DS}$ constitutes the postsynaptic activity, i.e., the response of the postsynaptic neuron (Fig. 5c-ii). We first assessed the short-term plasticity of our n-OPECT. Keeping a constant $V_{GS}$ while applying a light pulse generates a spike-like response on the $I_{DS}$, similar to the facilitation or potentiation of synaptic information (Supplementary Fig. 12a). Supplementary Fig. 12a shows that the retention time can be extended with higher light intensity, longer pulse duration, or higher $V_{GS}$. In an O$_2$-free electrolyte, charges generated upon pulsatile illumination remain within the film, thereby causing long-time retention (Supplementary Fig. 12b). Based on the kinetic differences between excitation and relaxation, we generated a spike-timing-dependent plasticity function by keeping the excitation pulse constant (100 ms, 600 mW/cm$^2$) and modulating the relaxation time in between pulses (Fig. 5c-iii). Neurons address action potentials ranging from sub-Hz to 200 Hz; here, the light cycle frequency spans from 0.2 Hz to 6.67 Hz. We observed that shorter presynaptic relaxation times (higher light cycle frequency) promote greater postsynaptic $I_{DS}$ (Fig. 5c-iv, Supplementary Fig. 12c). Additionally, the postsynaptic response is wavelength dependent, where the shorter the wavelength, the larger $I_{DS}$, due to the energy carried per photon being able to excite electronic states of higher energy[51]. The normalized current response versus light cycle frequency at three wavelengths is displayed in Fig. 5c-v. For a given light power and upon a frequency-dependent pulsing scheme, the n-OPECT possesses an RGB color recognition ability.

We investigated the photophysical properties and the light-induced charge generation mechanism of an n-type OMIEC film using electrochemical as well as ultrafast spectroscopic methods. Upon illumination, the film generated charged species; and when interfacing high-work function electrode, the presence of a Schottky barrier promoted a high carrier photogeneration rate and prolonged carrier lifetime. The generation of these charges resulted in a decrease in the surface potential of the film, and the aqueous electrolyte infiltrating the film further enhanced the efficiency of charge separation. When the film was utilized as the microscale gate and the channel of an OECT, the light-activated electrochemical potential and capacitance change of the n-type gate was effectively converted into a significant modulation of the n-type channel conductivity. The light-to-current transduction was achieved solely by the single n-type polymer film, without involving a Faradic process or necessitating the use of additional electron scavengers or donors. The photovoltage response was independent of oxygen presence and compatible with both aqueous and solid electrolytes. Furthermore, the illumination did not cause degradation of either the channel or the gate, and it resulted in a remarkable increase in OECT gain, reaching up to 40-fold. This configuration allowed for high sensitivity to even small changes in light intensity. We identified three potential areas for n-OPECT: as a photodetector of a PPG setup monitoring the heart rate, as a component of a light-triggered inverter circuit, and as an artificial synapse whose plasticity is tuned by light pulses. For the latter, the hysteresis of the light excitation/relaxation dynamics was leveraged to mimic the spike-timing-dependent plasticity of the artificial synapse using light stimuli of different frequencies and wavelengths (RGB). As light can activate the transistor switch, the n-OPECTs work with low power demand. When used for biosensing, the dramatically increased gain with light can allow the detection of very low concentrations of biomarkers or weak physiological signals. The fact that the gate electrode can be addressed with light alone, the electrolyte can be any medium-bearing mobile ions, and the n-type gate can be combined with other channel materials allows building this device with various form factors and

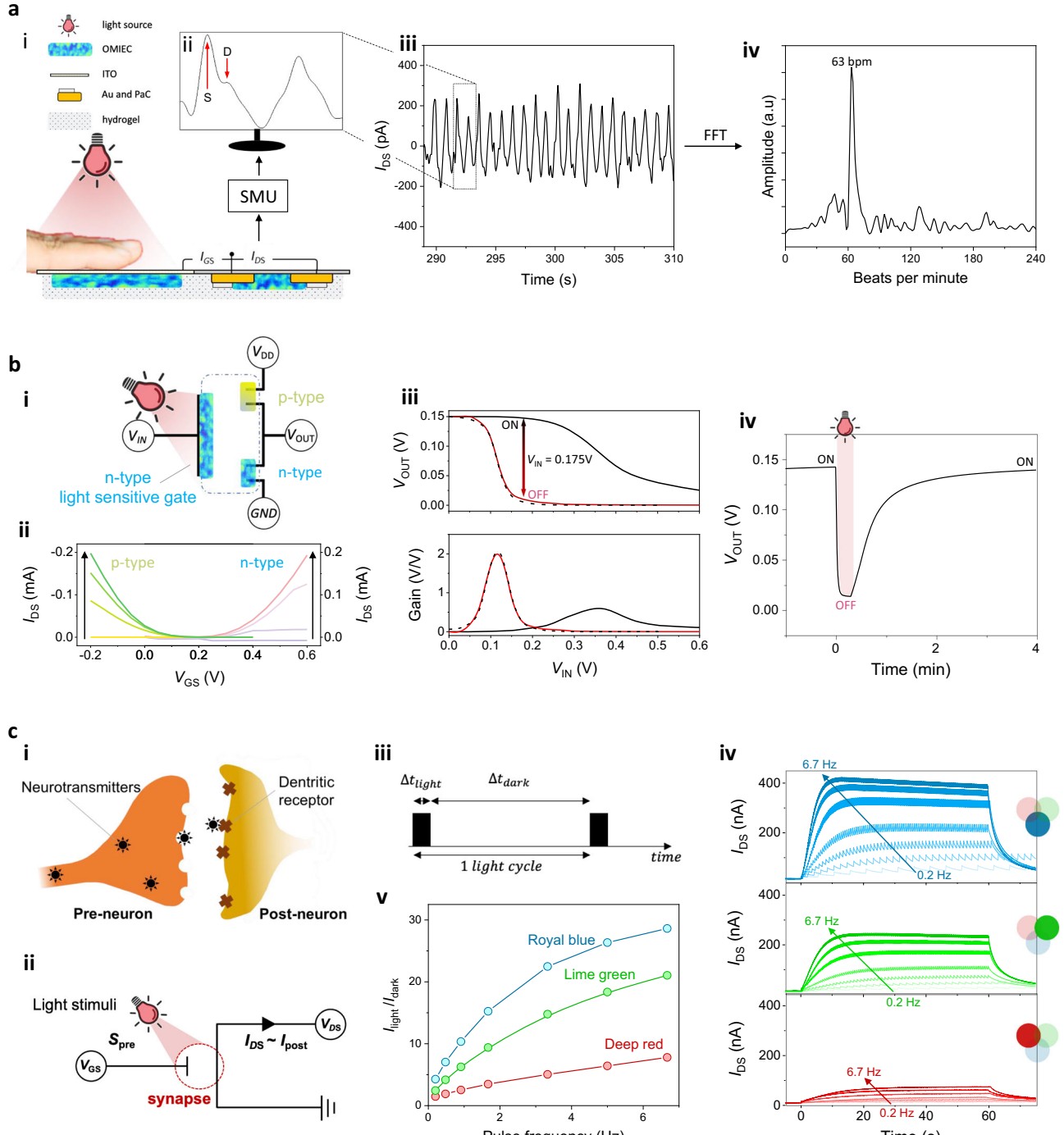

**Fig. 5 | n-OPECT applications. a-i** Schematic of the PPG platform, not to scale. The p(C6NDI-T) was coated on an ITO substrate. $V_{DS} = V_{GS} = 0.4$ V. **a-ii** $I_{DS}$ captured in one cycle showing the systolic and diastolic features. **a-iii** The $I_{DS}$ vs. time. **a-iv** Fast Fourier transform of the data in (iii), highlighting the dominant frequency of 63 bpm. **b-i** Schematic representation of the OPECT inverter. Input voltage ($V_{IN}$) is applied to the light-sensitive n-type gate. The n and p channels in series are connected to the ground (GND) and the driving voltage ($V_{DD}$), respectively. The output voltage ($V_{OUT}$) is the voltage point between the two channels. PBS (blue dotted line) connects the two channels and the common gate. **b-ii** Transfer curves of the p-type channel (left, $V_{DS}$ scan from 0 to −0.15 V) and of the n-type channel (right, $V_{DS}$ scan from 0 to 0.15 V). **b-iii** Output voltage ($V_{OUT}$, top) and gain (bottom) vs. input voltage ($V_{IN}$) in the dark (solid black line), under red light (solid red line), and in the dark (Ag/AgCl pellet as the gate electrode, dotted black line). **b-iv** Output voltage

vs. time at $V_{IN} = 0.175$ V. The light was switched on at t = 0 s and kept on for 20 s; **c-i** Schematic representation of a biological synapse **c-ii** The synaptic n-OPECT where $V_{GS}$ and light stimuli are the presynaptic cues ($S_{pre}$). Postsynaptic information ($I_{post}$) is reflected in $I_{DS}$. **c-iii** The light cycle profile. The black block represents the light pulse (600 mW/cm²) applied for 100 ms ($\triangle t_{light}$) followed by the relaxation in dark, the duration of which was varied ($\triangle t_{dark}$, see Supplementary Table 3). **c-iv** Real-time $I_{DS}$ response to light cycles applied at different frequencies using royal blue (top), lime green (middle), and deep red (bottom). Light cycles were switched ON at t = 0 s and lasted for 60 s (120 s for the two lowest frequencies). Arrows represent the evolution of the channel current with an increase in light cycle frequency. $V_{DS} = V_{GS} = 0.4$ V. **c-v** Normalized current response vs. pulse frequency of light at various wavelengths.

performance metrics, broadening the application areas beyond what is demonstrated in this work.

## Methods

### Materials

The photosensitive p(C$_6$NDI-T)[27] and the p(g0T2-g6T2)[49] were synthesized as previously reported. The phosphate-buffered saline solution (PBS) was prepared following the manufacturer's instructions (Merck). It contained about 137 mM NaCl, 2.7 mM KCl, 10 mM Na$_2$HPO$_4$, and 1.8 mM KH$_2$PO$_4$.

### OECT fabrication

OECTs were fabricated on 4-inch glass wafers. The wafers were cleaned using a piranha bath (H$_2$O$_2$:H$_2$SO$_4$, ratio 1:4), washed with water, and cleaned with O$_2$ plasma (Nanoplas DSB 6000). The OECT channels, pads, and interconnects were defined using standard photo-lithography steps. To perform the lift-off step, the wafers were coated with a photoresist bilayer consisting of LOR 5B (Microchem) and S1813 (Shipley) and exposed to UV light using the EVG 6200 mask alignment system and developed using MF319 developer. A 10 nm layer of Cr and a 100 nm layer of Au were deposited using magnetron sputtering (Equipment Support Company Ltd. ESCRD4) and lifted using appropriate solvents. After the lift-off step, the first Parylene C layer was deposited with a thickness of 1.7 μm using an SCS Labcoater 2 with silane as an adhesion promoter. A second Parylene C layer was vaporized to act as the sacrificial layer for polymer film patterning. A layer of AZ9260 was spun cast and developed using AZ developer as a mask for reactive ion etching (Oxford Instruments Plasmalab 100–ICP 380), which was used to expose the channels and pads for polymer deposition. The channels were 10 μm in length and 100 μm in width, whereas the lateral Au electrode had an area of 500 × 500 μm$^2$. p(C$_6$NDI-T) and p(g0T2-g6T2) solutions (5 mg/mL) were prepared by dissolving the polymers in chloroform. These solutions were spin-cast on the OECT active area at 800 rpm for 45 s with an acceleration of 400 rpm/s.

### Light source

We used a DC2200 LED Driver (ThorLabs) operating ThorLabs mounted LEDs for the light irradiation. The different ThorLab light sources we used are as follows: Royal Blue LED (M455L4), Deep Red LED (M660L4), Far Red LED (M730L5), and Infra-Red LED (M850L3). The light source calibration, and conversion from LED current input (from the driver controller) to light intensity output, was performed with the digital optical power and energy meter (PM100D, ThorLabs) combined with the slim photodiode power sensor (S130VC, ThorLabs). The distance between the OMIEC (or photodiode during calibration) and the light source was maintained at 1 cm throughout the study.

### Quartz crystal microbalance with dissipation monitoring (QCM-D)

QCM-D measurements were performed with a Q-sense analyzer (QE401, Biolin Scientific) using gold-coated crystal sensors. The QCM-D response of bare sensors was monitored in air and 0.1 M NaCl. Fluid injection into the chamber causes large changes in the QCM-D signals, which must be excluded from the mass uptake calculations. The films were spin-coated onto previously measured clean crystal sensors. We compared the absolute difference among several overtones between the bare and the coated sensors using the "Stitch data" function of the Q-Soft software. Mass uptake was calculated using the Sauerbrey equation, which relates the changes in mass to the frequency differences, using one overtone (n) as described in detail in other reports[28].

### Kelvin probe force microscopy (KPFM)

Scans were obtained with a Veeco Dimension 3100 Scanning Probe System operated with amplitude modulation KPFM (AM-KPFM)

detection technique. We fixed the lift height to 20 nm and the drive amplitude to 500 mV. We used SCM-Pit-V2 probes commercialized by Bruker (Nominal Resonant Frequency: 10 KHz, Spring Constant: 2.8 N/m). The film was coated on ITO substrate; the spin-coating parameters were 1500 rpm with an acceleration of 800 rpm/s. The surface was illuminated using the 660 nm deep red LED commercialized by ThorLabs and operated at 10 mA (4.2 mW/cm$^2$). When determining the work function of the different surfaces with and without light, we used the formula: $\varphi_{sample} = \varphi_{tip} - e.V_{CPD}$, with φ, the work function in eV, e being the elementary charge and $V_{CPD}$ is the contact potential difference measured in volt. The tip work function ($\varphi_{tip}$) was first determined by sampling the blank ITO substrate reported with a work function of 4.7 eV. Light illumination came from the bottom (ITO side). Gwyddion software was used for analysis and data post-treatment.

### Transient absorption (TA) spectroscopy

Transient absorption (TA) spectroscopy was carried out using a home-built pump-probe setup. The output of a Ti:sapphire amplifier (Coherent LEGEND DUO, 800 nm, 4.5 mJ, 3 kHz, 100 fs) was split into three beams (2 mJ, 1 mJ, and 1.5 mJ). Two of them were used to separately pump two optical parametric amplifiers (OPA; Light Conversion TOPAS Prime). The 2 mJ TOPAS generates wavelength-tunable pump pulses (240–2600 nm, using Light Conversion NIRUVIS extension), while the 1 mJ TOPAS generates signal and idler only (1160–2600 nm). The pump wavelength was fixed at 660 nm, except for the resonant excitation experiment in Supplementary Fig. 8. A fraction of the 1.5 mJ output of the Ti:sapphire amplifier was focused into a c-cut 3 mm thick sapphire window, thereby generating a white light supercontinuum from 500 to 1600 nm. The pump-probe delay time was achieved by varying the pump path length using a broadband retroreflector mounted on a 600 mm automated mechanical delay stage (Newport linear stage IMS600CCHA controlled by a Newport XPS motion controller), generating delays from −400 ps to 8 ns. Pump and probe beams were focused on the sample to spot sizes of 0.84 mm and 0.09 mm diameter (from a Gaussian fit at 86.5% intensity), as measured using a beam profiler (Coherent LaserCam-HR II). The samples were kept under a dynamic vacuum of <10$^{-5}$ mbar, and pump and probe beams were incident on the substrate side of the sample (i.e., closer to the ITO/polymer interface). The transmitted fraction of the white light was guided to a custom-made prism spectrograph (Entwicklungsbüro Stresing), where it was dispersed by a prism onto a 512-pixel CMOS linear image sensor (Hamamatsu G11608-512A). The probe pulse repetition rate was 3 kHz, the excitation pulses were mechanically chopped to 1.5 kHz, and the detector array was read out at 3 kHz. Adjacent diode readings corresponding to the transmission of the sample after excitation and in the absence of an excitation pulse were used to calculate ΔT/T. Measurements were averaged over several thousand shots to obtain a good signal-to-noise ratio. The chirp induced by the transmissive optics was corrected with a home-built Matlab code.

### Time-resolved fluorescence spectroscopy

Samples were excited using the second harmonic of a wavelength-tunable Ti:sapphire oscillator (Coherent CHAMELEON ULTRA I, 690–1040 nm, 2.9 W, 80 MHz, 140 fs). The excitation wavelength was fixed to 400 nm, and the beam was routed to the sample with a spot diameter of 0.93 mm and fluence of 15.6 nJ/cm$^2$ (i.e., irradiance of 1.25 W/cm$^2$). Samples were kept under a dynamic vacuum of <10$^{-5}$ mbar, and the excitation beam was incident on the substrate side of the sample (i.e., closer to the ITO/polymer interface). The fluorescence of the samples was collected by an optical telescope (consisting of two plano-convex lenses) and focused onto the slit of a spectrograph (Princeton Instruments Acton Spectra Pro SP2300) and detected with a streak camera (Hamamatsu C10910) system. The instrument

response function (IRF) of the streak camera using the synchroscan unit (M10911) was 9.86 ps. The data was acquired using the streak camera software (HPDTA) in photon counting mode.

## Doping-induced absorption spectroscopy

Molecular doping was used to generate negative charges using a strong one-electron dopant Cobaltocene. The neat film was prepared by spin-coating p($C_6$NDI-T) solution in chloroform on a pre-cleaned quartz substrate; subsequently, the dopant was spin-coated on the top. Cobaltocene was dissolved in an orthogonal solvent isopropanol (IPA), with a concentration of (0.3 mg/mL) to create a bilayer without dissolving the p($C_6$NDI-T) film underneath.

To characterize the doped film, UV-Vis absorption spectra of the film were measured before and after doping. Upon doping, the absorbance of the p($C_6$NDI-T) film changes, and this change can be calculated using the differential transmission formula as in equation 1 below:

$$\frac{\Delta T}{T} = 10^{-\Delta A} - 1 = \left( \frac{T_{doped} - T_{neat}}{T_{neat}} \right) \tag{1}$$

Where $T_{neat}$ and $T_{doped}$ are the transmittances of the neat and doped film, and the resulting spectrum is assigned to anion.

## Cyclic voltammetry and electrochemical impedance spectroscopy (EIS)

We used a three-electrode setup with the p($C_6$NDI-T) coated Au electrode as the working electrode, platinum (Pt) wire as the counter electrode, and Ag/AgCl as the reference electrode. The p($C_6$NDI-T) was cast on an Au-pattern circular electrode presenting a diameter of 500 µm. Cyclic voltammograms were recorded at 25 mV/s by sweeping the potential on the working electrode from 0.25 V to -0.8 V vs. Ag/AgCl using a VSP 300 BioLogic Science Instrument potentiostat. None of the first CV cycles were used as they differed from the successive ones. Electrochemical impedance spectra were recorded using the same instrument with a 10 mV modulation amplitude over a frequency range from 100 kHz to 0.1 Hz and over various DC offsets applied versus the Ag/AgCl reference electrode.

## IDE and OECT measurements

We recorded the current-voltage characteristics of our devices using a Keithley 2602 A dual source meter. We monitored the real-time changes in the channel current of the OECT at a constant $V_{DS}$ and a constant $V_{GS}$. After a steady baseline was obtained for the current, different light illuminations irradiated the platforms. The limit of detection was determined using three times the average noise signal amplitude of the baseline current.

## Determination of the electrochemical potentials of coated electrodes, IDE, and OECT terminals

All electrochemical potentials are determined using a VSP 300 Bio-Logic Science Instrument potentiostat VSP 300 BioLogic Science Instrument potentiostat utilizing the standard configuration presenting an input impedance of 1 TOhm and 25 pF capacitance. To determine the IDE and OECT terminal potentials while operating the device at desired biasing conditions. The general setup is the same as reported by Druet et al.[29]. Two potentiostat channels applied the potentials to the drain and the gate with respect to the source. The other three potentiostat channels were used to monitor the potential change of the gate, source, and drain with respect to the Ag/AgCl reference electrode.

## Controlled atmosphere/electrolyte experiments

$O_2$-free experiments were carried out in a glove box (MBRAUN, UNIlab pro Box) atmosphere where the $O_2$ concentration was always below 10 ppm. To conduct $O_2$-free experiments in the wet state, $N_2$ was purged into 1x PBS for at least 45 min before bringing the solution inside the glove box.

## Spectroelectrochemistry studies

Measurements were performed using an Ocean Optics HL-2000-FHSA halogen light source, QP600-1-SR-BX optical fibers, and Ocean Optics QE65 Pro Spectrometer. The OceanView software was first calibrated using a blank ITO substrate placed in the cuvette filled with 1x PBS. The samples were electrochemically doped to the desired potential manually using PalmSens4 portable potentiostat (PreSens) in a three electrodes setup using an Ag/AgCl reference electrode, a Pt coil counter electrode, and the polymer-coated ITO electrode as the working electrode. Absorbance spectra were recorded using the Oceanview software.

## Photoplethysmography recordings

An agarose-based hydrogel comprising 1 %wt agarose in 1x PBS was placed on top of the device when liquid (T > ca. 40 °C) and let to cool down (ca. 4 °C) and solidify for 30 min. The agarose hydrogel electrolytically connected the gate to the channel. The finger was pressed against the ITO gate electrode backside, illuminated by the LED from the top. While measuring the pulsatility, we applied $V_{DS} = V_{GS} = 0.4$ V and illuminated the volunteer's finger with a deep red LED (660 nm, 150 mW/cm$^2$). The PPG signal from the n-OPECT was acquired using a Keithley 2602 A dual source meter. PPG raw data were first passed through a 10 Hz low-pass infinite impulse response (IIR) filter for high-frequency de-noising and then processed with a discrete wavelet transform method using a "sym8" wavelet for artifact removal in the software MATLAB[22]. We used as PPG reference device the Shimmer3 GSR+ Optical Pulse (iMotions, København, Denmark). The study protocol was thoroughly reviewed and approved by the KAUST Institutional Biosafety and Bioethics Committee (IBEC Project NO. 21IBEC040). One volunteer (male, 28 years old) participated in this study with informed consent acquired before conducting the experiments.

## Reporting summary

Further information on research design is available in the Nature Portfolio Reporting Summary linked to this article.

## Data availability

The data that support the findings of this study are available within the article, its Supplementary Information, or from the authors.

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

## Acknowledgements

The research reported in this publication was supported by funding from KAUST, Office of Sponsored Research (OSR), under award numbers REI/1/4204-01, REI/1/4229-01, OSR-2015-Sensors-2719, and OSR-2018-CRG7-3709. C.E.P. acknowledges support from the KAUST Global Fellowship Program under the auspice of the Vice President for Research. I.M. acknowledges financial support from KAUST OSR CRG10, by EU Horizon2020 grant agreement no. 952911, BOOSTER, grant agreement no. 862474, RoLA-FLEX, and grant agreement no. 101007084 CITYSOLAR, EPSRC Projects EP/T026219/1 and EP/W017091/1. The authors thank Dr. Helen Bristow, Dr. Maxime Babics, Dr. Ilke Uguz, Dr. Julien Gorenflot and Dr. Wenchao Yang for the fruitful discussions. The schematic in Fig. 1e was created by A. Bigio, a scientific illustrator at KAUST.

## Author contributions

V.D. fabricated the devices and performed experiments involving electrodes, OECTs and IDEs. A.K. and D.O. helped with the electrical characterization of devices. D.O. and J.S. performed QCM-D measurements. C.E.P., N.A., and F.L. performed and analyzed the TA spectroscopy, DIA spectroscopy, and TRF spectroscopy. V.D., Y.Z., and A.K. performed the PPG experiments. V.D., Y.Z., and J.S. performed the logic circuit experiments. V.D., L.S., and A.K. performed the artificial synapse experiment. L.A. assisted V.D. with the characterization of the two photodetectors. P.D.N. conducted the RDE experiments. S.G. and I.M. provided the p(C6NDI-T) and the p-type polymer. S.I. conceived the research, designed the experiments, and supervised the work. V.D. wrote the paper with S.I. All authors were involved in the discussion and participated in paper input.

## Competing interests

The authors declare no competing interests.
