## [Peer Review File · Nature Communications]

REVIEWERS' COMMENTS

Reviewer #2 (Remarks to the Author):

The R1 submitted by Inal and colleagues thoroughly addressed the original comments I had, and I believe the extra experiments done with different electrolytes / dielectric constants were valuable and interesting. I enthusiastically support publication in the present form.

Reviewer #3 (Remarks to the Author):

Druet et al. provided a revised version of their manuscript which appropriately addressed my comments and questions, and those of the other reviewers. I do not have further concerns. As such, I recommend publication.

Point-to-Point Responses

Reviewer #2 (Remarks to the Author):

The R1 submitted by Inal and colleagues thoroughly addressed the original comments I had, and I believe the extra experiments done with different electrolytes / dielectric constants were valuable and interesting. I enthusiastically support publication in the present form.

Reply: Thank you for the comment.

Action: No further action is required.

Reviewer #3 (Remarks to the Author):

Druet et al. provided a revised version of their manuscript which appropriately addressed my comments and questions, and those of the other reviewers. I do not have further concerns. As such, I recommend publication

Reply: Thank you for the comment.

Action: No further action is required.